# BPSK Circuit Based on SDC Memristor

**DOI:** 10.3390/mi13081306

**Published:** 2022-08-12

**Authors:** Ran Gao, Yiran Shen

**Affiliations:** Institute of Modern Circuits and Intelligent Information, Hangzhou Dianzi University, Hangzhou 310018, China

**Keywords:** SDC (self-directed channel) memristor, BPSK, modulation, demodulation

## Abstract

Digital communication based on memristors is a new field. The main principle is to construct a modulation and demodulation circuit by using the resistance variation characteristics of the memristor. Based on the establishment of the Knowm memristor simulation model, firstly, the modulation circuit is designed by using the polarity and symmetry of the memristor and combined with the commercial current feedback amplifier AD844. It is proved that the modulated signal based on the memristor is a strong function of phase, and the demodulation circuit is designed accordingly. All simulation circuits are based on the actual commercial physical device model. The analytical expression of the output signal of the modulation and demodulation circuit is deduced theoretically, and the communication performance of the whole system is simulated by LTSpice. At the same time, the influence of the parasitic capacitance of the memristor on the circuit performance is also considered. After the simulation verification, the hardware circuit experiment of the modulation and demodulation circuit is carried out. The waveforms of the modulated signal and the demodulated signal are measured by an oscilloscope. The experimental results are completely consistent with the simulation and theoretical results.

## 1. Introduction

The concept of memristor was put forward by Chua for the first time [1,2]. Until May and June 2008, three papers were published in a row in nature to report the discovery of memristor [3,4,5]: HP laboratories found that a two-layer titanium dioxide film sandwiched between two platinum sheets had the characteristics of the memristor. It was the first time confirming the physical existence of nano-memristors from theory and experiment, which caused great shock in the industry and academia, so memristors become a new, hot research field [6]. However, the memristor design by HP laboratories is still unavailable in the commercial market because of its fabrication complexity and high cost, which delays the commercialization of real-time applications of memristors. Recently, the Knowm company in the United States released the first commercial discrete memristor [7], which enables the academic community to carry out experiments on memristors in the laboratory.

Memristors can be applied to analog signal processing because of their rich dynamic characteristics [8]. Using the characteristic that memristance is a strong function of signal frequency and phase, a memristor can even be applied to the field of digital communication. At present, there are relevant reports. For example, the memristor is proposed to be used as spectrum enhancement (frequency multiplier, modulator, mixer, etc.) in [9]. Recently, attempts to use memristor characteristics in communication have been reported, such as a UWB receiver [10] and a memristor-based modulator [9,11,12,13,14]. Ref. [12] proposed an amplitude modulation (AM) circuit based on the linear doping drift model. In addition, frequency shift keying (FSK), amplitude modulation (AM), binary phase shift keying (BPSK), and amplitude shift keying (ASK) modulators are also proposed in [11]. In [13], the simulation circuit of the memristor element is experimentally studied and used in the new design of FSK, ASK, and phase shift keying (PSK) modulators. In addition, when designing the new structure of the compression sensing system, the random modulator based on a memristor is used, and the storage and switching characteristics of memristor devices are used [14]. The architecture of memory reference receivers based on memristors is introduced, in which the operation is performed by calculating the correlation between reference waveform and received signal [10]. Refs. [15,16] summarized some of the above work and presented, as an application, a modulation and demodulation scheme together with a complete transceiver based on memristors.

However, the modulation or demodulation circuit designed in the published literature is either based on the memristor emulator or the ideal linear or nonlinear memristor model. There is still a distance from the actual physical circuit based on the memristor, so it is impossible to carry out hardware verification.

Given the above shortcomings, this paper proposes a complete modulation/demodulation transceiver link for the BPSK communication system based on the actual commercial memristor model and carries out the experimental verification of the hardware circuit.

This paper is organized as follows. In Section 2, the model of the commercial Knowm memristor is proposed and programmed with LTSpice. It is worth noting that, unlike the traditional deterministic model of memristor, this model introduces random variables, so it has a wider range of applications. Section 3 designs the modulation circuit based on this model, and Section 4 designs the modulation circuit. These two parts not only give a detailed circuit system diagram but also clearly explain the working principle of the circuit. Section 5 is the simulation and experimental results.

## 2. Modeling and Simulation of Knowm Memristor

Different from the traditional deterministic memristor model, random variables can be used to model the memristor. The specific idea is to regard the memristor as a collection of conductive channels, which have different resistance values, and the channel itself can be composed of nanoparticles such as ions. By applying an external voltage, the channel is switched between ON and OFF to change the resistance of the device. With the increase in the number of channels, the memconductance will be larger and larger, which is because the device has more conductive paths. A series of memristors can be modeled by modifying the number of channels. Ref. [17] gives the model of Knowm memristor as follows:(1)$$\left\{\begin{array}{l} dx/dt=\frac{1}{\tau }[\frac{1}{1+e^{-\beta (v-V_{ON})}}(1-x)-(1-\frac{1}{1+e^{-\beta (v+V_{OFF})}})\cdot x]\\ i=v\cdot g\\ g=\frac{x}{R_{ON}}+\frac{1-x}{R_{OFF}} \end{array}\right.$$
where *x* is the state variable, that is, the number of normalized conductive channels, and $\beta =V_{T}{}^{-1}$ is the reciprocal of the thermal voltage. The other parameters are shown in Table 1.

It should be noted that, although $V_{ON}$ and $V_{OFF}$ have the same value (0.27 V), they have different physical meanings and may have different values in modeling some types of memristors.

Figure 1 is the time-domain transient simulation circuit built after incorporating the Knowm memristor model in the LTSpice. Figure 2 and Figure 3 are the simulation waveforms obtained from this.

## 3. Modulation Circuit

BPSK is the simplest form of PSK (phase shift keying). It is a digital modulation scheme, which realizes data modulation by changing the phase of the carrier. BPSK uses two phases 180° apart to represent “1” and “0”. It is suitable for low-cost passive transmitters and has the best anti-interference performance. In addition, it is also widely used in wireless local area networks (LAN), radio frequency identification (RFID), and Bluetooth communication.

Figure 4 shows the BPSK modulation circuit diagram, and Table 2 shows the parameters in the diagram. This circuit scheme is not new and directly cited from [13]; however, we use the real memristor model instead of the ideal model mentioned in [13]. In the circuit, three AD844 chips are connected as shown in the figure. The AD844 chip is the current feedback operational amplifier (CFOA) of the AD company. This type of op-amp provides a closed-loop bandwidth that is determined primarily by the feedback resistor and is almost independent of the closed-loop gain. The AD844 is free from the slew rate limitations inherent in traditional op-amps and other current-feedback op-amps. It combines high bandwidth and very fast large signal response with excellent DC performance. Although optimized for use in current-to-voltage applications and as an inverting mode amplifier, it is also suitable for use in many noninverting applications. The AD844 can be used in place of traditional op-amps, but its current feedback architecture results in much better AC performance, high linearity, and an exceptionally clean pulse response. The power supply voltage of AD844 is ±15 V, AD844 chip terminal *z* is the current output terminal, and terminal *v* is the voltage output terminal. The z-terminal output current *i* of CFOA1 copies the input current of the amplifier, and then *i* is split into *i*_1_ and *i*_2_. The currents *i*_1_ and *i*_2_ provide a bias current for memristors M1 and M2 through CFOA2 and CFOA3, respectively, and generates a voltage on it, which is output to terminal *v* of CFOA2 and CFOA3, and then superimposed at the output node to form output voltage Vo.

The operating principle of the circuit is explained as follows: the circuit based on the current feedback operational amplifier (CFOA1) is used to convert the unipolar binary digital input signal into the current. The output current of CFOA1 is distributed between two equal resistors *R*_2_ and *R*_3_, which in turn provide bias for the two memristors *M*_1_ and *M*_2_ connected to CFOA2 and CFOA3 terminal z.

If the current flowing through the resistor is $i_{R_{1}}$, then
(2)$$i_{R_{1}}=\frac{V_{in}-2.5}{R_{1}}$$

$i_{R_{1}}$ is precisely copied at the z-terminal of CFOA1, so $i=i_{R_{1}}$. The carrier voltage at the in-phase terminal of CFOA2 is 6sin(2πft)V, and we have $R_{2}i_{1}+R_{3}i_{2}=6\sin(2\pi ft)$. Considering $R_{2}=R_{3}$, the following is obtained by using Kirchhoff voltage and current law:(3)$$i_{1}+i_{2}=\frac{6\sin(2\pi ft)}{R_{2}}$$
(4)$$i_{2}-i_{1}=i=\frac{V_{in}-2.5}{R_{1}}$$

According to Equations (3) and (4), we have
(5)$$i_{1}=\frac{1}{2}(\frac{6\sin(2\pi ft)}{R_{2}}-\frac{V_{in}-2.5}{R_{1}})$$
(6)$$i_{2}=\frac{1}{2}(\frac{6\sin(2\pi ft)}{R_{2}}+\frac{V_{in}-2.5}{R_{1}})$$
and we have the memristor equations:(7)$$\left\{\begin{array}{l} dx_{1}/dt=\frac{1}{\tau }[\frac{1}{1+e^{-\beta (V_{M_{1}}-V_{ON})}}(1-x_{1})-(1-\frac{1}{1+e^{-\beta (V_{M_{1}}+V_{OFF})}})\cdot x_{1}]\\ g_{1}=\frac{x_{1}}{R_{ON}}+\frac{1-x_{1}}{R_{OFF}}\\ V_{M_{1}}=i_{1}/g_{1} \end{array}\right.$$
(8)$$\left\{\begin{array}{l} dx_{2}/dt=\frac{1}{\tau }[\frac{1}{1+e^{-\beta (V_{M_{2}}-V_{ON})}}(1-x_{2})-(1-\frac{1}{1+e^{-\beta (V_{M_{2}}+V_{OFF})}})\cdot x_{2}]\\ g_{2}=\frac{x_{2}}{R_{ON}}+\frac{1-x_{2}}{R_{OFF}}\\ V_{M_{2}}=i_{2}/g_{2} \end{array}\right.$$

Because the polarity of the two memristors is opposite, one increases with the current flowing, and the other inevitably decreases with the current flowing. Therefore, one of the resistances of the two memristors is larger and another is smaller. When a sinusoidal carrier voltage is applied in CFOA2 in-phase terminal, the sinusoidal voltage drop generated on the two memristors will have opposite polarity and different amplitude, and the output voltage is the superposition of the voltage drop on the two memristors:(9)$$V_{o}=\frac{1}{2}(V_{M_{1}}+V_{M_{2}})$$

Therefore, according to the binary digital input of the CFOA1 in-phase terminal, the output voltage will have two opposite phases.

## 4. Demodulation Circuit

From the perspective of analytical expression, firstly, the memristor state equation:(10)$$dx/dt=\frac{1}{\tau }[\frac{1}{1+e^{-\beta (v(t)-V_{ON})}}(1-x)-(1-\frac{1}{1+e^{-\beta (v(t)+V_{OFF})}})\cdot x]$$
is a first-order differential equation, rearranged and we obtain:(11)$$dx/dt+\frac{1}{\tau }(1+\frac{1}{1+e^{-\beta (v(t)-V_{ON})}}-\frac{1}{1+e^{-\beta (v(t)+V_{OFF})}})\cdot x=\frac{1}{\tau }\frac{1}{1+e^{-\beta (v(t)-V_{ON})}}$$

We set
(12)$$\left\{\begin{array}{l} P(t)=\frac{1}{\tau }(1+\frac{1}{1+e^{-\beta (v(t)-V_{ON})}}-\frac{1}{1+e^{-\beta (v(t)+V_{OFF})}})\\ Q(t)=\frac{1}{\tau }\frac{1}{1+e^{-\beta (v(t)-V_{ON})}} \end{array}\right.$$

Then, the solution of Equation (11) is
(13)$$x=e^{-\int P(t)dt}[\int Q(t)e^{\int P(t)dt}dt+C]$$
where *C* is an arbitrary constant, which is determined by the initial resistance of the memristor. Since *v*(*t*) is a function of phase, *x* is also a function of phase, which can be recorded as *x*(*θ*), where *θ* is a phase. So according to Equation (1), there are
(14)$$g(\theta )=\frac{x(\theta )}{R_{ON}}+\frac{1-x(\theta )}{R_{OFF}}$$

Thus, the memconductance is also a function of phase.

Figure 5 reflects the relationship between the input signal phase and the memristor conductance.

From 0 to 1 ms, the signal has a 0 phase and from 1 to 2 ms, it has a 180-degree phase. It can be seen that the memristor conductance is a strong function of the input signal phase.

Figure 6a,b shows the system block diagram and circuit diagram of the proposed BPSK demodulator, respectively, and Table 3 shows the relevant parameter values. Although the topology of this circuit is not new and can be found in [13], we performed a modification here, inverting the amplifier and adopting the real memristor model instead of the ideal model in [13].

The circuit principle is explained as follows: as shown in Figure 6b, the BPSK demodulating signal is applied to the voltage divider between the memristor element and the resistor. The output of the voltage divider is
(15)$$V_{1}=V_{in}R_{1}/(R_{1}+R_{m})$$

As mentioned earlier, the memristance *R_m_* is a strong function of the phase of the input signal. Since *R_m_* varies with the input phase, the amplitude of *V*_1_ also varies with the phase. In other words, the voltage divider converts the change of the sinusoidal input phase into the change of sinusoidal output amplitude, thereby creating a PSK to ASK (amplitude shift keying) converter based on the memristor. The envelope detector generates a DC signal proportional to the ASK signal amplitude after the voltage divider in Figure 6b. Resistor *R*_2_ creates a discharge path to automatically reset the output of the envelope detector at the beginning of each bit without resetting the circuit, thereby creating a multilevel output based on the phase of the input signal. The power consumed through the discharge path can be determined by the difference between the energy stored in the capacitor during the high level and the low level dividing the discharge time, thus $0.5C_{1}(V_{h}{}^{2}-V_{l}{}^{2})/t_{d}$, where *V_h_* is the capacitor voltage during bit “1”, *V_l_* is the capacitor voltage during bit “0”, and *t_d_* is the discharge time.

The circuit does not need any carrier recovery circuit, which reduces the related overhead in other traditional BPSK demodulators and means it is suitable for ultra-low power applications. In addition, the demodulator has the characteristics of simple structure, small volume, and low power consumption. It can be used in the monolithic integrated design of wireless implantable neural recording systems.

Figure 6b shows the BPSK demodulator circuit starting from the memristor resistor voltage divider. It converts different amplitude currents related to different phases of the received signal into different amplitude voltages through the series resistance (*R*_1_). The inverting amplifier is used to amplify the output voltage of the voltage divider so that it exceeds the forward voltage of the diode. If the output level of the voltage divider is high, it cannot be used. Then, there is a passive simple envelope detector for detecting the signal peak, and finally, a first-order low-pass filter (LPF) to reduce the ripple at the output.

Generally, in BPSK, bit “1” is represented by several sinusoidal periods with a 0° phase shift, while a bit “0” is represented by several sinusoidal periods with a 180° phase shift. Therefore, the corresponding memresistance of bit “1” is low in bit time (as shown in Figure 5). The voltage drop at both ends of *R*_1_ is sinusoidal, and the peak value is relatively large. Therefore, the output of the envelope detector is high, indicating the voltage level of logic “1”. On the contrary, when a 180° phase-shifted sinusoidal waveform is applied, the average value of *R*_m_ is larger in the case of bit “0”. Therefore, the voltage drop at both ends of *R*_1_ is sinusoidal, the peak value is relatively small, and the output of the envelop detector corresponds to the voltage level of bit “0”.

This is a very simple BPSK demodulator, which does not need expensive hardware and avoids the use of carrier recovery circuits. It should be noted that the values of *C*_1_ and *R*_2_ must be carefully selected to ensure that the conversion time (the time required for the peak detector to convert between different levels) is sufficiently less than the bit duration and greater than the carrier period.
(16)$$T_{c}<\tau <T_{b}$$

*T_c_* and *T_b_* are carrier period and bit duration, respectively, and *τ* is the time constant of capacitor-discharge, *τ* = *R*_2_*C*_1_.

Finally, the output signal of Figure 6b is sent to the waveform shaping circuit based on the comparator of Figure 6c to obtain the modified demodulated signal. The power supply voltage of LM 339 is ±15V. The reason that we need the comparator is that the rising edge of the signal $V_{out1}$ out of the LPF is not steep enough because of the nonideality of the simple RC LPF, but by using the comparator, the steepness can be improved, and a clearer pulse shape is obtained.

## 5. Simulation and Experiment Results

In this section, SPICE simulation experiments are carried out to verify the modulation/demodulation circuit. Figure 7a shows the transient time-domain simulation circuit diagram of the modulation/demodulation. In the simulation experiment, firstly, the memristor model is written as SPICE code, and then the circuit symbol is automatically generated by LTSpice, that is, the block in the figure. The upper sub-circuit of the circuit diagram is a modulation circuit, and its output is output1. Then, output1 is directly sent to the lower sub-circuit of the circuit diagram, that is, the demodulation circuit and its output are output2. Then, output2 is sent to the input of the comparator circuit, and the demodulated signal output3 is obtained from its output. The simulation results are shown in Figure 7b. The upper waveform is a modulated signal. The high bit is 5 V, representing bit “1”, and the low bit is 0 V, representing bit “0”; the middle waveform is the modulated signal, i.e., output1, and the lower waveform is the demodulated signal, i.e., output3.

Due to the parasitic capacitance effect of the memristor, we paralleled the parasitic capacitance of 10 nF at both ends of the memristor in the circuit shown in Figure 7a, and the simulated waveform is shown in Figure 8. It can be observed that the pulse width of the output demodulation waveform is widened, which is caused by the parasitic capacitor charging and discharging and the capacitor tends to maintain the potential across its terminals for a certain time, that is, the RC product tends to be larger, and the charging and discharging time tends to be longer.

The hardware experiment setup is shown in Figure 9a, and the required instruments are a regulated power supply, a signal generator, and an oscilloscope. The regulated power supply provides the supply voltage of three AD844s and one LM339. The signal generator uses two channels to generate a modulating signal square wave and carrier sine wave, respectively. Figure 9b shows the modulated waveform, which is different from the simulation waveform because the memristor model used in the simulation is different from the actual device and cannot fully simulate the behavior of the real device. However, we still observe the sudden change of the 180° phase, which is a significant feature of the BPSK signal. Figure 9c shows the demodulation waveform. The interested reader may refer to [7] for further information about real Knowm SDC memristor device fabrication and electrical characteristics.

## 6. Conclusions

This paper presents the BPSK modulation circuit and demodulation circuit based on a commercial Knowm memristor. Based on the simulation model of the Knowm memristor, the modulation circuit is designed by using the polarity and symmetry of the memristor and commercial current feedback amplifier AD844. In all cases, the unipolar binary digital input is first converted into a current. This current is used to bias the memristor by changing its resistance. Thus, exploiting the inherent memristor characteristic of having two values of resistance; that is *R_ON_* and *R_OFF_*. It is proved that the modulated signal based on the memristor is a strong function of phase, and the receiving and demodulation circuit is designed. The designed transceiver circuits employ the commercial passive memristor and hence are implementable as hardware prototypes and also have low power. For demonstration purposes, we use a 1 kHz carrier frequency; however, it can be easily extended to several MHz. The modulation/demodulation circuit can be further extended to the modulation and demodulation of ASK and FSK signals.

Compared with related work in [11,13,15,16], we use real commercial memristor products. Beyond simulation, we also carried out hardware experiments and measured the modulated and demodulated waveforms. The proposed memristor model is suitable for analog as well as digital applications such as analog computations, neuromorphic circuits, adaptive filters, and signal processing circuits. Because of its low power, it is especially appropriate for low-power biomedical applications.

## Figures and Tables

**Figure 1 micromachines-13-01306-f001:**
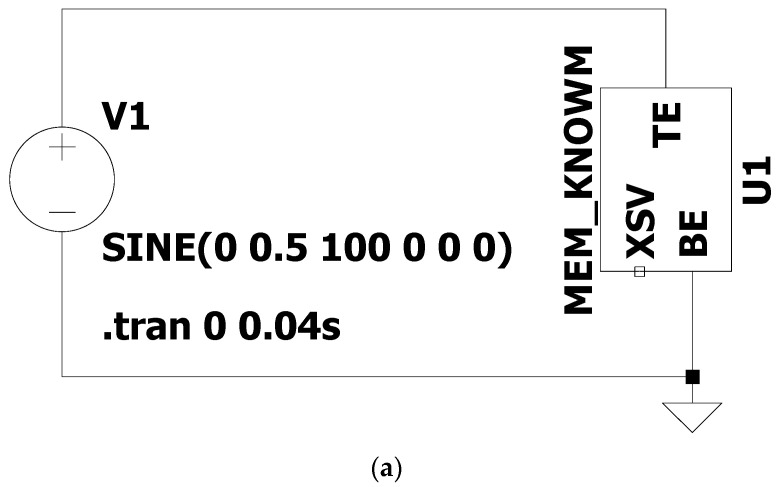
Memristor simulation circuit, MEM_ Knowm is the model of Knowm memristor, and XSV is the state variable. (**a**) Schematic (**b**) Knowm memristor model.

**Figure 2 micromachines-13-01306-f002:**
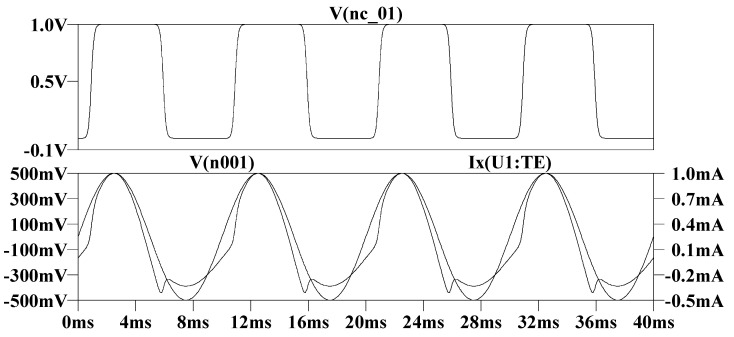
Time domain simulation results: the curve above is the relationship between state variables and time, the sine curve below is the input voltage waveform, and the other curve is the current waveform.

**Figure 3 micromachines-13-01306-f003:**
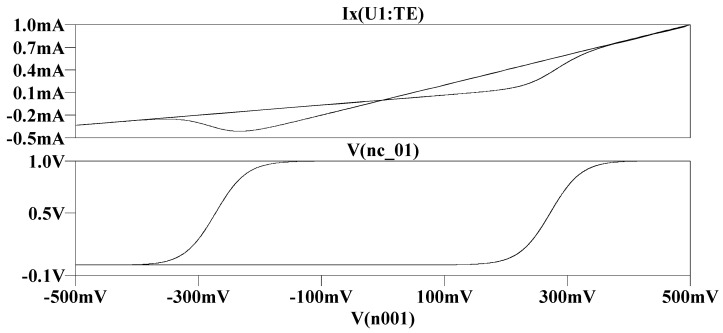
The upper figure shows the hysteresis curve of memristor input voltage and response current, and the lower figure shows the hysteresis curve of memristor input voltage and state variable.

**Figure 4 micromachines-13-01306-f004:**
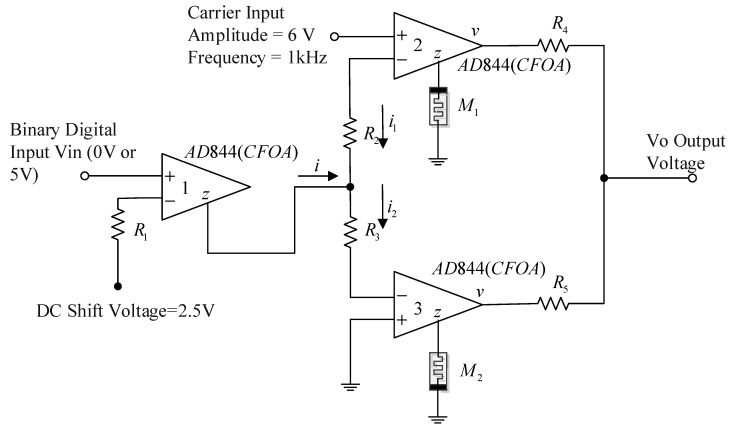
BPSK modulation circuit.

**Figure 5 micromachines-13-01306-f005:**
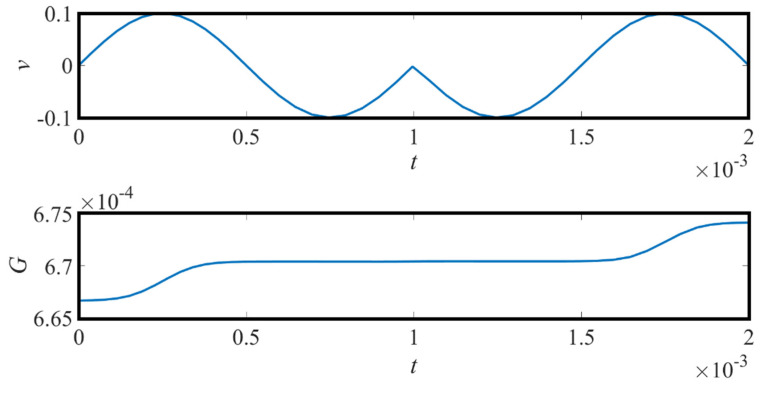
Memristor conductance is a strong function of the input signal phase.

**Figure 6 micromachines-13-01306-f006:**
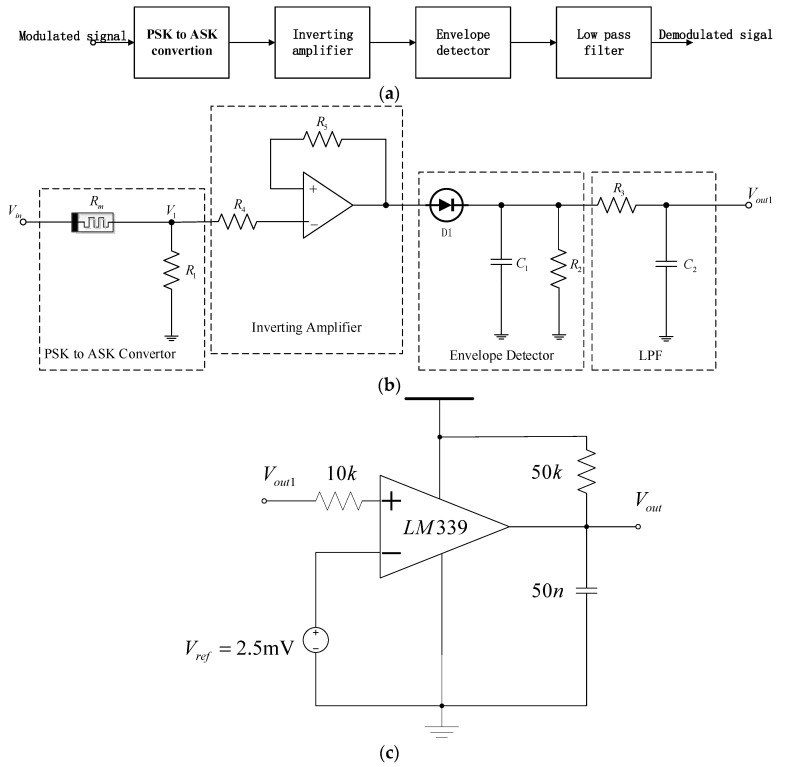
Demodulator system block diagram and circuit diagram. (**a**) BPSK demodulator system block diagram (**b**) BPSK demodulator circuit diagram (**c**) Comparator circuit.

**Figure 7 micromachines-13-01306-f007:**
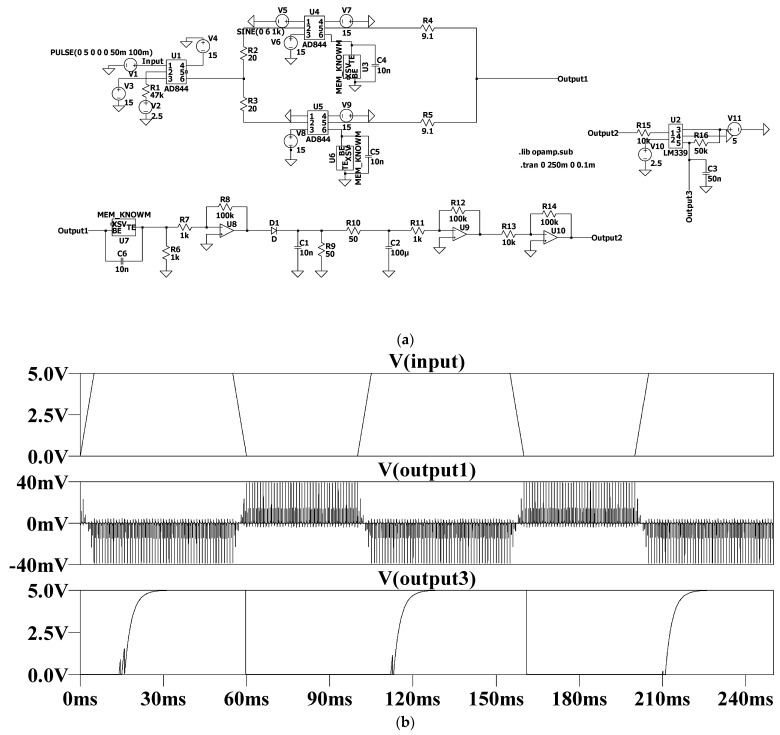
Simulation circuit and waveform. (**a**) Time domain simulation circuit diagram of modulator/demodulator. (**b**) The simulation results.

**Figure 8 micromachines-13-01306-f008:**
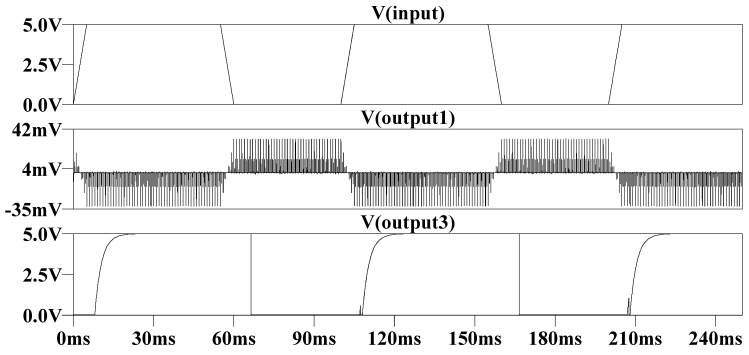
Waveform after considering parasitic capacitance effect.

**Figure 9 micromachines-13-01306-f009:**
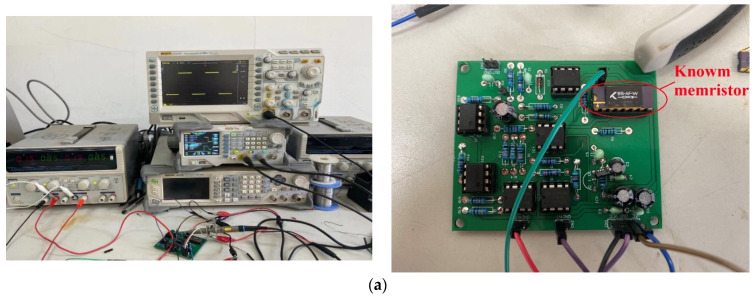
Hardware experiment setup and waveform. (**a**) Experimental setup (**b**) Modulated waveform (**c**) Demodulated waveform.

**Table 1 micromachines-13-01306-t001:** List of parameters of the memristor model.

Parameter	Meaning	Reference Value (S.I. Units)
$R_{ON}$	Minimum resistance	500
$R_{OFF}$	Maximum resistance	1500
$V_{ON}$	ON threshold voltage	0.27
$V_{OFF}$	OFF threshold voltage	0.27
τ	time constant	0.0001
T	temperature	298.5

**Table 2 micromachines-13-01306-t002:** List of parameters of Figure 4.

Parameter	Value (Ω)
*R* _1_	47 k
*R* _2_	20
*R* _3_	20
*R* _4_	9.1
*R* _5_	9.1

**Table 3 micromachines-13-01306-t003:** List of parameters of Figure 6.

Parameter	Value (S.I Units)
*R* _1_	1 kΩ
*R* _2_	50 Ω
*R* _3_	50 Ω
*R* _4_	1 kΩ
*R* _5_	100 kΩ
*C* _1_	10 nF
*C* _2_	100 μF

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
