# Peer review of "BPSK Circuit Based on SDC Memristor"

_micromachines, 2022, doi:10.3390/mi13081306_

Round 1
Reviewer 1 Report
The manuscript (micromachines-1838433), BPSK circuit based on SDC memristor, shows an interesting result by using memristor device for BPSK modulation circuit. Authors presented comprehensive analysis in this work. However, some technical comments still need to be addressed before further confirm the outcomes, shown as following - 1. Authors mentioned that "Figure 9 (b) shows the modulated waveform, which is different from the simulation waveform because the memristor model used in the simulation is different from the actual device and can not fully simulate the behavior of the real device.", can authors shows what is the basically device characterizations in this memristor? What are the device structures here and what type of the memristor referred to here? This is quite important since it should provide the detailed info why the actual device behaviors are different from the simulated devices behaviors. Please provide the further information below - a. What is the forming voltage and activation process? b. What is the switching voltage and energy? DC and AC characterizations. c. What is the fabricated device structures and layer thickness information? d. What are the endurance and retention characterizations? e. In the experimental setup, where is the memristor and other device components? Please provide the layout of the connection in detail and mark components in detail. f. Please comment on what would be the major difference between the actual device characterizations and simulated device behaviors. Due to the above comments, this referee would like to put the manuscript status as "Major Revision" in the current phase.Author Response
Please see the attachment.

Reviewer 2 Report
The manuscript “BPSK circuit based on SDC memristor” presents very interesting mamristors applications for binary phase shifting key modulators. It has the great application character for many devices and involving memristors can decrease the power consumption of the device, what seems to be a key here.
The manuscript has a good shape where the reader is introduced to the topic from theory through simulations to the physical realization. The last part has great scientific values, as this confirms that the proposed design is suitable.
Authors have used the memristor fabricated by the Knowm company. They provide the Self-Directed-Channel (SDC) memristors with different dopants (W, C, Cr, Sn). Depending on the dopant kind, the Knowm’s memristors have different characteristics (switching energy and timings, voltage threshold etc.). It would be worth to provide the information which memristors have been considered in the circuit presented and why.
Authors also considered in the simulations the parasitic capacitance. The capacitor of 10nF seems to be high. Please elaborate about this parasitic aspect. Why authors decided to pick that capacitor value and how it impacts to the simulations comparing to the measurement of the real design. How the parasitic capacitor has been implements in the spice model described in the manuscript.
The simulations have been performed using the LTSpice environment. I would recommend if the spice code of the memristor element model would be also shared, this might be of interests of other readers and definitely might impact the citation factor.
There are also some typos noticed e.g. line 25 word Three should be three (small letter), similarly in line 29 So->so etc. Please consider rewriting the sentence in the second paragraph of the Introduction chapter “The dynamic…, which makes memristor can…” (line 36-38). Line 84 should be together with 85. Also, there are missing subscripts in the variables e.g. V1 -> V_1 (line 196), Rm->R_m (line 196) and so on (l. 201, 206, 223, 235,236 ).
The figures from LTSpice simulation are very hard to read due to the font size, please increase. Please consider to change background into white (LTSpice has this functionality).
Considering the above, I recommend the manuscript to be published in Micromachines magazine.
Round 2
Reviewer 1 Report
Authors have replied to this referee in detail. No further comments from this referee.